# Antimicrobial Properties of Amino-Acid-Derived N-Heterocyclic Carbene Silver Complexes

**DOI:** 10.3390/pharmaceutics14040748

**Published:** 2022-03-30

**Authors:** Adrián Sánchez, Carlos J. Carrasco, Francisco Montilla, Eleuterio Álvarez, Agustín Galindo, María Pérez-Aranda, Eloísa Pajuelo, Ana Alcudia

**Affiliations:** 1Departamento de Química Inorgánica, Facultad de Química, Universidad de Sevilla, Aptdo 1203, 41071 Sevilla, Spain; ash03whitetiger@gmail.com (A.S.); ccarrasco1@us.es (C.J.C.); galindo@us.es (A.G.); 2Instituto de Investigaciones Químicas, CSIC-Universidad de Sevilla, Avda. Américo Vespucio 49, 41092 Sevilla, Spain; ealvarez@iiq.csic.es; 3Departamento de Química Orgánica y Farmacéutica, Universidad de Sevilla, 41012 Sevilla, Spain; mariapar89@gmail.com; 4Departamento de Microbiología y Parasitología, Universidad de Sevilla, 41012 Sevilla, Spain; epajuelo@us.es

**Keywords:** silver complexes, chiral, antimicrobial, imidazolium-carboxylate, amino acid

## Abstract

Complexes {Ag[NHC^Mes,R^]}_n_ (R = H, **2a**; Me, **2b** and **2b’**; *^i^*Pr, **2c**; *^i^*Bu, **2d**), were prepared by treatment of imidazolium precursor compounds [Im^Mes,R^] (2-(3-mesityl-1*H*-imidazol-3-ium-1-yl)acetate, **1a**, (*S*)-2-*alkyl*(3-mesityl-1*H*-imidazol-3-ium-1-yl)acetate, **1b**–**d**, and (*R*)-2-methyl(3-mesityl-1*H*-imidazol-3-ium-1-yl)acetate, **1b’**, with Ag_2_O under appropriate conditions. They were characterised by analytical, spectroscopic (IR, ^1^H, and ^13^C NMR and polarimetry), and X-ray methods (**2a**). In the solid state, **2a** is a one-dimensional coordination polymer, in which the silver(I) cation is bonded to the carbene ligand and to the carboxylate group of a symmetry-related Ag[NHC^Mes,H^] moiety. The coordination environment of the silver centre is well described by the DFT study of the dimeric model {Ag[NHC^Mes,H^]}_2_. The antimicrobial properties of these complexes were evaluated versus Gram-negative bacteria *E. coli* and *P. aeruginosa*. From the observed MIC and MBC values (minimal inhibitory concentration and minimal bactericidal concentration, respectively), complex **2b’** showed the best antimicrobial properties (eutomer), which were significantly better than those of its enantiomeric derivative **2b** (distomer). Additionally, analysis of MIC and MBC values of **2a**–**d** reveal a clear structure–antimicrobial effect relationship. Antimicrobial activity decreases when the steric properties of the R alkyl group in {Ag[NHC^Mes,R^]}_n_ increase.

## 1. Introduction

The antimicrobial properties of silver have been known since ancient times, a use that continued until the middle of the past century, when antibiotics (e.g., penicillin) replaced silver in most bacterial infection clinical treatments [1]. However, the development of antimicrobial resistance to most common antibiotics led to a resurgence in the use of silver as a biocide agent that exhibits low toxicity to humans. Therefore, during recent decades, metallic silver, silver(I) simple salts, and silver(I) complexes have long been used as antimicrobial agents in various medical applications, including dental treatments, catheters, and healing of burn wounds [2,3]. The antimicrobial activity of silver is attributed to the bioavailability of the Ag(I) cation, which depends on the delivery methods, solubility, or presence of ligands [4]. Although the mechanisms of action of silver cations are not yet fully understood, it is thought that antibacterial toxicity is associated with its affinity for soft bases, such as the thiols groups (R-SH), that are found in the enzymes or proteins of the bacterial membrane. Other evidence suggests that the presence of silver ions disrupts the activity of the bacterial electron transport chain and can also lead to the production of reactive oxygen species (ROS) and the depletion of antioxidants into cells [5,6].

Silver(I)-N-Heterocyclic Carbene (Ag(I)-NHC) complexes have shown strong antimicrobial properties against many Gram-negative and Gram-positive bacterial strains [7,8,9,10,11,12,13,14,15]. As an interesting example, Ag(I)-NHC complexes derived from caffeine were successfully evaluated against a panel of bacterial strains with a special emphasis on resistant respiratory pathogens, showing very low genotoxicity, which is interesting for clinical use [9,11]. The strong coordination ability of NHCs with silver gives rise to stable complexes that can slowly release silver ions, thus retaining the antimicrobial effect over a longer period of time. Furthermore, the substitution of the NHC ligand could be used to modify the silver’s release and control its delivery into the cell [8].

On the other hand, the use of α-amino acids for the synthesis of ligands is a simple strategy to generate bioinorganic metal systems [16,17]. Particularly, interest in chiral imidazole-type or imidazolium-type carboxylate compounds has increased markedly [18,19,20]. They are easily synthesised from amino acids and are convenient starting precursors for the preparation of enantiopure substrates, such as ionic liquids [21] and NHC ligands [22,23,24,25,26,27]. Additionally, they have been used as useful bridging ligands, acting as linkers in the construction of homochiral coordination polymers or metal–organic frameworks [28,29,30,31,32] and as chirality inductors in asymmetric catalysis [33,34,35].

Following our recent interest in the study of the antimicrobial properties of Ag(I)-complexes with imidazolium-dicarboxilate ligands based on α-amino acids [36], the antimicrobial activity of several Ag(I)-NHC complexes obtained from α-amino acids-based imidazolium-monocarboxylate precursors was explored. The presence of the α-amino acid in the NHC ligand provides interesting characteristics to the silver complex, such as chirality and the biocompatibility required for its possible therapeutic application. Here, we describe the synthesis and characterisation of {Ag[NHC^Mes,R^]}_n_ complexes, which were prepared by treating Ag_2_O with the corresponding imidazolium precursor, [Im^Mes,R^] [37,38,39]. The biocide effectiveness of these complexes was tested by performing microbiological assays to determine the minimal inhibitory concentrations (MIC) and minimal bactericidal concentrations (MBC) in Gram-positive bacteria (*Staphylococcus pseudintermedius y Staphylococcus aureus*) and Gram-negative bacteria (*Escherichia coli* and *Pseudomonas aeruginosa*) bacteria. Furthermore, their mechanisms of action and their effect on biofilm formation were also evaluated as a very important role in the development of antimicrobial resistance [40].

## 2. Materials and Methods

### 2.1. Materials

All synthetic preparations were carried out under a nitrogen atmosphere, while other operations were performed under aerobic conditions. Solvents were purified and dried appropriately prior to use, using standard procedures. The chemicals were obtained from commercial sources and used as supplied. Zwitterionic imidazolium compounds [Im^Mes,R^], **1**, were prepared according to the procedure reported by Mauduit et al. [20,37,38]. Infrared spectra were recorded on a PerkinElmer FT-IR Spectrum Two spectrophotometer (Waltham, MA, USA), in pressed KBr pellets or using the ATR technique. NMR spectra were recorded on Bruker AMX-300 or Avance III spectrometers (Billerica, MA, USA) at the *Centro de Investigaciones, Tecnología e Innovación* (CITIUS) of the University of Sevilla, with ^1^H and ^13^C{^1^H} NMR shifts referenced to the residual signals of deuterated solvents. All data are reported in ppm downfield from Si(CH_3_)_4_. Polarimetry was carried out using a JASCO P-2000 digital polarimeter and the measurements were made at room temperature (concentration of ca. 10 mg/mL). Elemental analyses (C, H, N) and high-resolution mass spectra were conducted by CITIUS of the University of Sevilla on an Elemental LECO CHNS 93 analyser (LECO Corporation, St. Joseph, MI, USA) and a QExactive Hybrid Quadrupole-Orbitrap mass spectrometer from Thermo Fischer Scientific (Waltham, MA, USA), respectively.

### 2.2. Synthesis of the {Ag[NHC^Mes,R^]}_n_ Complexes

Complexes **2b**–**d** have previously been described [37,38,39], but the experimental procedure we use varies slightly and is described here. Additionally, the complete spectroscopic and analytical characterisation of these Ag–NHC complexes is also included. The general synthetic method is described for **2a**, while for the other **2** complexes, only the scale and yield were included.

#### 2.2.1. Complex {Ag[NHC^Mes,H^]}_n_ (**2a**)

(*S*)-2-(3-mesityl-1*H*-imidazol-3-ium-1-yl)acetate, [Im^Mes,H^] (**1a**), (1.22 g, 5 mmol) was mixed with Ag_2_O (1.10 g, 5 mmol), suspended in dry CH_2_Cl_2_ (10 mL) and stirred for 18 h under a N_2_ atmosphere at room temperature. The resulting solution was filtered and concentrated, and then Et_2_O (1 mL) was added. Cooling to −20 °C afforded compound **2a** as a pale brown solid. Yield 1.12g (1.59 mmol, 64%). IR (cm^−1^): 3390 (m), 1605 (vs), 1490 (m), 1380 (vs), 1305 (s), 1240 (m), 1200 (w), 1145 (w), 1100 (w), 1035 (w), 970 (w), 935 (w), 850 (s), 795 (s), 735 (s), 698 (s), 585 (s). ^1^H RMN (CDCl_3_, 300 Hz, ppm): 1.73 (s, 6H, *o*-C*H*_3_), 2.27 (s, 3H, *p*-C*H*_3_), 4.76 (s, 2H, C*H*_2_), 6.66 (s, 1H, C*H*, C^4^/C^5^ im), 6.74 (s, 2H, *m*-C*H*), 7.54 (s, 1H, C*H*, C^4^/C^5^ im). ^13^C{^1^H} NMR (CDCl_3_, 75.47 Hz): 17.5, 17.7 (s, *o*-*C*H_3_), 21.1 (s, *p*-*C*H_3_), 65.84 (s, *C*H_2_), 122.2, 123.4 (s, *C*H, C^4^/C^5^ im), 129.5 (s, *m*-*C*), 134.6 (s, *o*-*C*), 135.1 (s, *p*-*C*), 135.8 (s, *o*-*C*), 138.8 (s, *i*-*C*), 172.1 (s, *C*=O), 181.2 (br s, C^2^ im). ESI-MS (negative mode): found *m*/*z* 351.0265 for [Ag[NHC^Mes,H^] + H]^+^, calculated *m*/*z* 351.0257 for C_14_H_16_AgN_2_O_2_^+^. Elemental Anal. Calc. for C_14_H_15_AgN_2_O_2_·1.4CH_2_Cl_2_: C, 39.35; H, 3.82; N, 5.96. Found: C, 39.46; H, 3.94; N, 5.84.

#### 2.2.2. Complexes {Ag[(*S*)-NHC^Mes,Me^]}_n_ (**2b**) and {Ag[(*R*)-NHC^Mes,Me^]}_n_ (**2b’**)

Compound (*S*)-2-(3-mesityl-1H-imidazol-3-ium-1-yl)propanoate, [(*S*)-Im^Mes,Me^] (**1b**) (0.129 g, 0.5 mmol). Yield 0.102g (0.14 mmol, 56%). IR (cm^−1^): 2920 (w), 1607 (vs), 1489 (m), 1456 (m), 1382 (s), 1349 (s), 1311 (m), 1263 (w), 1225 (s), 1166 (w), 1109 (w), 1079 (w), 1026 (w), 964 (w), 933 (w), 877 (w), 852 (m), 726 (s), 698 (m), 618 (w), 586 (m), 503 (m). ^1^H NMR (CDCl_3,_ 300 Hz): 1.74 (d, ^3^*J* = 7.2 Hz, 3 H, C*H*_3_CH), 1.97 (br s, 6H, *o*-C*H*_3_), 2.32 (s, 3H, *p*-C*H*_3_), 5.28 (m, 1H, CH_3_C*H*), 6.86 (s, 1H, C*H*, im C^4^/C^5^ im), 6.93 (s, 2H, *m*-C*H*), 7.49 (s, 1H, C*H*, C^4^/C^5^ im). ^13^C{^1^H} NMR (CDCl_3_, 75.47 Hz): 17.7, 17.9 (s, *o*-*C*H_3_), 20.9 (s, *C*H_3_CH), 21.0 (s, *p*-*C*H_3_), 62.2 (s, *C*HCH_3_), 120.4, 121.8 (s, C*H*, im), 129.1, 129.2 (s, *C*H, C^4^/C^5^ im), 134.9, 135.3 (s, *o*-*C*), 135.8 (s, *p*-*C*), 138.9 (s, *i*-*C**)*, 175.0 (s, C=O), 180.1 (br s, C^2^ im). ESI-MS (negative mode): found *m*/*z* 365.0423 for [Ag[NHC^Mes,Me^] + H]^+^, calculated *m*/*z* 365.0414 for C_15_H_18_AgN_2_O_2_^+^. Elemental Anal. Calc. for C_15_H_17_AgN_2_O_2_·0.7CH_2_Cl_2_: C, 44.41; H, 4.37; N, 6.60. Found: C, 44.60; H, 4.49; N, 6.71 %. [α]_D_ (21 °C) = 23.1 ± 2,5 (H_2_O). Following the same experimental method, the complex {Ag[(*R*)-NHC^Mes,Me^]_2_} (**2b’**) was prepared in 25% yield. Compound **2b’** showed identical IR, ^1^H and ^13^C{^1^H} NMR spectra as its enantiomer **2b**. [α]_D_ (21 °C) = −21.0 ± 2.3 (H_2_O).

#### 2.2.3. Complex {Ag[(*S*)-NHC^Mes,*i*Pr^]}_n_ (**2c**)

Compound (*S*)-2-(3-mesityl-1H-imidazol-3-ium-1-yl)-3-methylbutanoate, [(*S*)-Im^Mes,*i*Pr^] (**1c**) (0.140 g, 0.5 mmol). Yield 0.098 g (0.15 mmol, 30%). IR (cm^−1^): 3127 (w), 2985 (w), 2870 (w), 1607 (vs), 1488 (m), 1463 (w), 1407 (m), 1360 (s), 1309 (m), 1262 (w), 1214 (m), 1166 (w), 1105 (w), 1082 (w), 1032 (w), 969 (w), 931 (w), 851 (m), 752 (vs), 735 (s), 682 (w), 584 (w), 500 (w). ^1^H NMR (CDCl_3,_ 300 Hz): 0.99 (d, ^3^*J* = 7.2 Hz, 3 H, C*H*_3_CH), 1.96 (br s, 6H, *o*-C*H*_3_), 2.33 (s, 3H, *p*-C*H*_3_), 2.47 (m, 1H, CH_3_C*H*), 4.83 (m, 1H, C*H*CH), 6.89 (s, 1H, C*H*, C^4^/C^5^ im), 6.94 (s, 2H, *m*-C*H*), 7.77 (s, 1H, C*H*, C^4^/C^5^ im). ^13^C{^1^H} NMR (CDCl_3_, 75.47 Hz): 17.6, 17.9 (s, *o*-*C*H_3_), 19.0, 20.2 (s, *C*H_3_CH), 21.0 (s, *p*-*C*H_3_), 32.5 (s, *C*HCH_3_), 65,8 (br s, *C*HCH), 121.4, 121.8 (s, *C*H, C^4^/C^5^ im), 129.1, 129.3 (s, *m*-*C*H), 134.9 (s, *p*-*C*), 135.1, 135.7 (s, *o*-*C*), 139.0 (s, i-*C**)*, 173.8 (s, C=O). ESI-MS (negative mode): found *m*/*z* 393.0737 for [Ag[NHC^Mes,*i*Pr^] + H]^+^, calculated *m*/*z* 393.0727 for C_17_H_22_AgN_2_O_2_^+^. Elemental Anal. Calc. for C_17_H_22_AgN_2_O_2_·0.15CH_2_Cl_2_: C, 50.74; H, 5.29; N, 6.90. Found: C, 50.74; H, 5.44; N, 6.64%. [α]_D_ (21 °C) = 30.2 ± 3.4 (H_2_O).

#### 2.2.4. Complex {Ag[(*S*)-NHC^Mes,*i*Bu^]}_n_ (**2d**)

Compound (*S*)-2-(3-mesityl-1H-imidazol-3-ium-1-yl)-4-methylpentanoate, [(*S*)-Im^Mes,*i*Bu^] (**1d**) was used (0.151 g, 0.5 mmol). Yield 0.106 g (0.13 mmol, 52%). IR (cm^−1^): 3110 (w), 3060 (m), 2870 (w), 1605 (vs), 1555 (m), 1540 (w), 1400 (m), 1360 (vs), 1300 (m), 1280 (w), 1235 (w), 1200 (s), 1155 (w), 1100 (m), 1020 (m), 970 (w), 920 (w), 890 (w), 850 (s), 740 (vs). ^1^H NMR (CDCl_3,_ 300 Hz): 0.88 (m, 6H, C*H*_3_CH), 1.26 (m, 1H, CH_3_*CH*), 1.86, 193 (s, 3H, *o*-C*H*_3_), 2.03 (m, 2H, CH*CH*_2_CH), 2.24 (s, 3H, *p*-C*H*_3_), 5.18 (br s, 1H, C*H*CH_2_), 6.79 (s, 1H, C*H*, C^4^/C^5^ im), 6.80 (s, 2H, *m*-C*H*), 7.34 (br s, 1H, C*H*, C^4^/C^5^ im). ^13^C{^1^H} NMR (CDCl_3_, 75.47 Hz): 17.6, 18.0 (s, *o*-*C*H_3_), 21.1 (s, *p*-*C*H_3_), 23.3, 25.0 (s, (*C*H_3_)_2_CH), 42.4 (sw, CH*C*H_2_CH), 65.8 (sw, *C*HCH_2_), 120.5, 122.06 (s, *C*H, C^4^/C^5^ im), 123.3 (s, *m-C*H), 129.1, 129.4 (s, *o-C*), 135.2 (s, *i*-C), 139.0 (s, *p*-C), 174.7 (s, C=O). ESI-MS (negative mode): found *m*/*z* 407.0881 for [Ag[NHC^Mes,*i*Bu^] + H]^+^, calculated *m*/*z* 407.0883 for C_18_H_24_AgN_2_O_2_^+^. Elemental Anal. Calc. for C_18_H_23_AgN_2_O_2_·0.4CH_2_Cl_2_: C, 50.09; H, 5.44; N, 6.35. Found: C, 50.36; H, 5.44; N, 6.42%. [α]_D_ (21 °C) = 27.3 ± 1.3 (H_2_O).

### 2.3. Single-Crystal X-ray Crystallography

A summary of the crystallographic data and the structure refinement results for compound **2a** is given in Appendix A (ESI). Crystals of suitable size for X-ray diffraction analysis were coated with dry perfluoropolyether, mounted on glass fibres and fixed in a cold nitrogen stream (T = 213 K) to the goniometer head. Data collection was carried out on a Bruker-AXS, D8 QUEST ECO, PHOTON II area detector diffractometer, using monochromatic radiation *λ*(Mo K_α_) = 0.71073 Å, by means of ω and φ scans with a width of 0.50 degrees. The data were reduced (SAINT [41]) and corrected for absorption effects by the multi-scan method (SADABS) [42]. The structures were solved by direct methods (SIR-2002 [43]) and refined against all *F*^2^ data by full-matrix least-squares techniques (SHELXTL-2018/3 [44]), minimising *w*[*F*_o_^2^
*− F*_c_^2^]^2^. All non-hydrogen atoms were refined anisotropically. Hydrogen atoms were included from calculated positions and refined by riding on their respective carbon atoms with isotropic displacement parameters. A search for solvent accessible voids for **2a** using *SQUEEZE* [45] showed two small volumes of potential solvents of 342 Å^3^ for each (a 121 electron count for each), whose solvent content could not be identified or refined with the most severe restraints. The corresponding CIF data represent *SQUEEZE*-treated structures with the solvent molecules handling a diffuse contribution of the overall scattering, and with the specific atom positions excluded from the structural model. The *SQUEEZE* results were appended to the CIF. CCDC 2143256 (**2a**) contains the Appendix A for this paper. The data can be obtained free of charge via: https://www.ccdc.cam.ac.uk/structures/.

### 2.4. Antimicrobial Studies and Microbiological Assays

#### 2.4.1. Bacterial Strains and Culture Conditions

Four bacterial strains were selected, two Gram-positive staphylococci, a *Staphylococcus pseudintermedius* LMG 22,219 and a *Staphylococcus aureus* CECT 5190, retrieved from the Belgium Coordinate Collection of Microorganisms and the Spanish Culture Collection (University of Valencia), respectively, and two Gram-negative strains, *Escherichia coli* CECT 434 and *Pseudomonas aeruginosa* CECT 110, both retrieved from the Spanish Culture Collection. All strains were inoculated on plates of tryptone soy agar (TSA) to isolate individual colonies. For conservation, liquid cultures in tryptone soy broth (TSB) with glycerol up to 15% (*v*:*v*) were frozen at −76 °C.

#### 2.4.2. Determination of Minimal Inhibitory Growth Concentrations (MIC) and Minimal Bactericidal Concentrations (MBC)

For the determination of the MIC, stock solutions of the **2** complexes were prepared at a concentration of 5 mM using CH_2_Cl_2_. Then, serial base two logarithmic dilutions of this 5 mM solution were prepared in Müeller-Hinton broth (MH), down to 0.00975 mM. The assay was performed in 96-well microtiter plates. After that, 200 μL of each solution were placed in the wells (in triplicate for each concentration). The control well contained MH medium and CH_2_Cl_2_ in the same proportion as previously used. Furthermore, cultures of *S. pseudintermedius* or *S. aureus,* as well as *E. coli* and *P. aeruginosa*, were grown overnight in TSB medium at 37 °C and 200 rpm from the previous day. The initial optical density of the cultures at 600 nm was determined and adjusted to 1.0 with sterile TSB. After that, all the wells of the microtiter plates were inoculated with 5 μL of the cultures for each of the strains, except the controls. The plates were sealed and incubated at 37 °C for 24 h. The MIC for each bacterium towards both compounds was determined after visual turbidity valoration, and also by measuring the optical density at 600 nm on a microtiter plate reader ASYS UVM340 [46]. For the MBC determination, 100 μL of the well content in the absence of turbidity was spread on tryptone soy agar (TSA) plates and then were incubated for 24 h at 37 °C. The MBC corresponded to the concentration that did not show development of any bacterial colonies [46]. In addition, a comparative study was conducted between the bacterial sensitivity ranges towards AgNO_3_, considered as the compound of reference for its antimicrobial properties, and **2**, using the techniques to determine MIC and MBC described above. Finally, in order to determine whether precursor ligands **1**, used in the synthesis of **2**, had antimicrobial activity per se, MIC assays were also performed.

#### 2.4.3. Determination of Antioxidant Enzymes and Thiobarbituric Acid Reactive Substances (TBARs)

The bacteria strains were grown in 50 mL of TSB medium for 24 h at 37 °C and 200 rpm, and were separated in five aliquots of 10 mL. The first one was maintained in TSB as a control, and for the other four, the **2** compounds at 1 × MIC and 1 × MBC were added for each strain, respectively. After that, they were incubated for another 24 h at 37 °C and 200 rpm. Moreover, the cultures were centrifuged at 8000 rpm for 5 min, the supernatants were dropped, and the bacterial pellets were resuspended in 2.5 mL of extraction buffer composed of 50 mM potassium phosphate, with pH 7.0 and containing 2 mM EDTA. In addition, the bacterial suspensions were sonicated for 3 periods of 30 s, separated by periods of 1 min to refrigerate the crude extract, using a sonicator Ultrasonic Processor (Hielscher) with amplitude 100% and cycle 0.8. All the procedures were performed in an ice bath. Subsequently, the homogenates were centrifuged at 10,000× *g* for 10 min at 4 °C, and the supernatants were transferred to clean tubes to be used for enzyme determinations. All enzymatic assays were performed at room temperature. Catalase (CAT) activity was measured by determining the disappearance of H_2_O_2_ at 240 nm (ε = 39.4 mM^−1^·cm^−1^) using a Perkin Elmer Lambda 25 UV/Vis spectrophotometer (Shelton, CT, USA) [47]. The reaction mixture was carried out in a quartz cuvette containing 800 μL of 50 mM sodium phosphate buffer (pH 7.6), 0.1 mM EDTA, and 100 μL of 3% H_2_O_2_. The addition of 100 μL of crude extract started the reaction, and the decrease in the OD at 240 nm was registered for 2 min.

The activity of the total peroxidases was determined by following the oxidation of pyrogallol to purpurogallin caused by H_2_O_2_ measured at 420 nm [48]. The reaction was carried out in 3 mL spectrophotometer cuvettes and contained 1.5 mL of 10 mM potassium phosphate buffer with pH 6.0, 0.1 mL of freshly made 0.4 M pyrogallol, and 0.1–0.2 mL enzyme extract. The reaction was started by adding 0.15 mL of freshly made 0.3% (*v*/*v*) H_2_O_2_. After an incubation period of 5 min at room temperature, the absorbance at 420 nm was measured. For the calculation of activity, a control without enzyme extract was prepared, and the basal oxidation of pyrogallol followed for 5 min. The increase in absorbance as a result of the basal oxidation of pyrogallol was detracted from the increase in absorbance of the samples. Activity was calculated using a molar extinction coefficient of ε = 12 mM^−1^ cm^−1^.

Nitroblue tetrazolium (NBT) photoreduction assay was performed in the presence of riboflavine at 560 nm to determine the superoxide dismutase (SOD) activity [49]. A 30 mL stock solution was prepared containing: 27.5 mL of potassium phosphate buffer (pH 7.6), 1 mL of 0.2 M EDTA, 1 mL of freshly prepared 1.5 mM NBT, and 0.5 mL of Triton X-100. The reaction cuvettes contained a total volume of 3 mL, composed of 2.88 mL of stock solution and 20 μL of crude extract. The blank reaction (to follow the basal photoreduction of NBT) was prepared without the crude extract. The reaction was started by adding 0.1 mL of 0.12 mM riboflavin to each cuvette. In addition, the cuvettes were exposed to white light (commercial fluorescent lights) for 5 min, and after that, the absorbance at 560 nm was determined. One unit of SOD activity was considered as the amount of enzyme that can inhibit 50% of NBT photoreduction by riboflavin [49].

For the determination of reactive substances of thiobarbituric acid (TBARS) standing out from malondialdehyde (MDA), bacterial strains were grown under the same conditions described previously (in the presence of 1 × MIC or 1 × MBC for each strain). Cultures were centrifuged at 8000 rpm for 5 min, and the pellets were resuspended in 3 mL of a mixture of 20% trichloroacetic acid containing 0.5% thiobarbituric acid [50]. After that, the homogenate was heated at 95 °C for 30 min followed by rapid cooling in an ice bath, and then centrifuged at 8000× *g* for 5 min. The concentration of MDA was calculated from the absorbance value at 532 nm measured with a PerkinElmer Lambda 25 UV/Vis spectrophotometer (Shelton, CT, USA), using the molar extinction coefficient ε = 155 mM^−1^·cm^−1^. Moreover, the protein content of crude extracts was determined using Bradford’s method [51], according to a calibration curve using bovine serum albumin (fraction V, Sigma) as a standard.

#### 2.4.4. Effects on Biofilm Formation: Evaluation by Colorimetric Technique and by Scanning Electron Microscopy (SEM)

The effect of the complexes on biofilm formation was evaluated using 96-well microtiter plates. A volume of 200 μL TSB medium (controls) or TSB medium supplemented with the **2** complexes in a concentration range from 5 mM to 0.0097 mM was placed in triplicate in the wells. After that, the wells were inoculated with 5 μL of overnight cultures, whose optical density at 600 nm was previously determined and adjusted to 1.0 with sterile TSB. Control wells containing an increasing concentration of either of the complexes were not inoculated. Finally, the microtiter plate was sealed and incubated at 28 °C for 24 h. The plate was then emptied, the wells were washed three times with 200 μL of distilled water, and the plate was dried at room temperature. After that, the biofilm formed was stained with 200 μL of crystal violet (1% *w*:*v*) in each well for 15 min. After washing the plate three times with water, 200 μL of a solution of acetic acid:ethanol (33%:67% *v*:*v*) was added to each well and the plate was incubated at room temperature for 30 min. Finally, the absorbance at 570 nm was measured [52].

In addition, the formation of biofilms on a glass surface was evaluated by SEM. For this purpose, 24-well polystyrene plates were used. Circular glass slides of 1 cm diameter were placed in the well fund. Then, a volume of 2 mL of TSB medium (controls) or TSB medium supplemented with silver **2** compounds at 1 × MIC and 1 × MBC for each strain was added to the well fund (sufficient volume to cover the glass slide). The wells were then inoculated with 100 μL of overnight grown cultures of each bacterial strain (optical density adjusted to 1.0). Plates were sealed and incubated at 28 °C for 4 days to allow biofilm formation. After this time, the glass slides were carefully transferred to a clean plate, washed three times with sterile distilled water, and then allowed to dry at room temperature. A volume of 5 mL of a mixture of 2.5% glutaraldehyde prepared in 0.2 M cacodilate buffer with pH 7.2 was poured into each well for 3 h at room temperature to fix the bacteria. After washing three times with 5 mL of 0.2 M cacodilate buffer pH 7.2, the samples were dehydrated in acetone series (50% to 100%) and dried using a critical point drier Leica EM CPD300 (Leica, Barcelona, Spain) at 31 °C and 73.8 bar. Dried samples were sputtered with Au-Pd (10 nm) and observed with a scanning electron microscope Zeiss EVO LS15 (Carl Zeiss Iberia, Madrid, Spain).

### 2.5. Computational Details

The electronic structure and geometries of [Im^Mes,R^], **1**, their carbene isomers, **1’**, and the [NHC^Mes,R^]^−^ ligands were investigated by using density functional theory at the B3LYP level [53,54] with the 6-311++G** basis set. This combination of method and basis set provides a good structural description of these type of compounds, according to the comparison of the structural parameters of the optimised structure of [HIm^Mes,*i*Bu^]^+^ cation with that of the reported crystal structure of [HIm^Mes,*i*Bu^]Cl (CSD refcode SOXWIN) [20]. For the silver complexes, the Ag atom was described with the LANL2DZ basis set [55], while other atoms were described with the 6-31G* basis set. Frequency calculations were carried out at the same theoretical level to identify all the stationary points as minima (zero imaginary frequencies) and to provide the thermal correction to free energies at 298.15 K and 1 atm. Molecular geometries were optimised without symmetry restrictions. DFT calculations were performed using the Gaussian 09 suite of programs (Gaussian, Inc., Wallingford, CT, USA) [56]. The coordinates of optimised compounds are reported in Appendix A.

## 3. Results and Discussion

### 3.1. Synthesis and Characterisation of the {Ag[NHC^Mes,R^]}_n_ Complexes

Complexes {Ag[NHC^Mes,R^]}_n_, **2**, were prepared by reaction of imidazolium precursors [Im^Mes,R^], **1**, with Ag_2_O in dried dichloromethane and were obtained as light brown solids with good yields (Figure 1). Complexes **2b**–**d** have previously been isolated as intermediate reagents in transmetalation reactions, but their characterisation was poorly described, and the experimental procedure we used varies slightly [37,38,39]. Antisymmetric COO vibrations of the carboxylate group appeared as a broad band at around 1605 cm^−1^ in the IR spectra of **2**, while bands within the 1380–1360 cm^−1^ range were assigned to symmetric COO vibrations. Besides the expected signals for the alkyl and aryl side chains, the ^1^H NMR spectra were characterised by resonances in the 6.66–6.89 and 7.34–7.77 ppm ranges for the two non-equivalent H atoms at the 4 and 5 positions in the imidazole ring, respectively. ^13^C{^1^H} NMR spectra give rise to a characteristic broad low-field resonance signal for the carbenic carbon at around 180 ppm (that signal was not observed in complexes **2c** and **2d**). Furthermore, carbon atoms of carboxylate groups resonate at around 175 ppm. The specific chirality of the synthesised complexes was determined using polarimetry. The specific [α]_D_ rotation of complexes **2b**–**d** with ligands **1b**–**d** from the natural α-amino acids *L*-alanine, *L*-valine, and *L*-leucine, respectively, is positive, while that of complex **2b’**, prepared from *D*-alanine, has the opposite sign and is similar in absolute value to its enantiomer **2b**.

### 3.2. X-ray Characterisation of Complex {Ag[NHC^Mes,H^]}_n_, ***2a***

For the **2** complexes, a 1:1 ratio of silver cations and NHC ligands was deduced from the elemental microanalysis. Furthermore, the ESI-MS spectrum showed fragmentation patterns that were consistent with the formation of monomeric Ag[NHC^Mes,R^] complexes; in all cases, the molecular ion displayed an appropriate isotopic ratio with the base peak at *m/z* 351.0265, 365.0423, 393.0737, and 407.0881 for complexes **2a**–**2d**, respectively. However, binuclear [38] and polymeric [37,39] formulations have been previously proposed for these complexes, and their particular solid-state structure is actually unknown. For this reason, we decided to identify this species by X-ray crystallography. Complex **2a** crystallises in the monoclinic system in the centrosymmetric *P*2**_1_**/c space group. The asymmetric unit is made up of the Ag[NHC^Mes,H^] entity (Figure 1a), in which the Ag(I) centre is coordinated with the carbene ligand (2.046(8) Å). The carboxylate group of NHC^Mes,H^ is bonded to a symmetry-related silver ion (2.100(6) Å), forming a C(1)-Ag(1)-O(1A) angle of 170.4(3)° (Figure 1b). This affords a coordination polymer that grows along the crystallographic *b* axis (Figure 1c and Appendix A). The carboxylate group is asymmetric (O(1)-C(14) 1.25(1) and O(2)-C(14) 1.19(1) Å) and in agreement with the observed κ^1^-O coordination mode. Other structural parameters are collected in Appendix A (ESI) and do not require further comments. Crystal packing in **2a** is achieved by non-classical C(sp^3^)-H⋯O hydrogen bonds between the methyl groups of mesityl belonging to one 1D chain and the oxygen atoms of the carboxylate groups of the neighbouring 1D chain (Appendix A). Within the unit cell, four interactions of this type are detected with C-H⋯O hydrogen-oxygen distances around 2.67 Å (see Appendix A).

There are a good number of NHC-carbene silver complexes with a coordinated carboxylate functionality as co-ligands. For these NHC–Ag–OC(O)R complexes, mononuclear compounds are structurally characterised by the expected linear C–Ag–O angle (range 163–179°, mean value of 172°) and the distances around the mean values of 2.12 Å for Ag–O (range 2.06–2.20 Å) and 2.07 Å for Ag–C (range 2.02–2.10 Å) [57]. Complex **2a** fits well within these parameters, with the exception of the Ag–O bond length (2.046(8) Å), which is slightly shorter than the found range. On the contrary, the Ag–C distance (2.100(6) Å) is in the longest part of the observed range. **2a** is the first example of NHC–Ag–OC(O)R complexes in which the carboxylate group comes from the coordinated carbene ligand, thus generating its polymeric nature. This fact is probably the reason for the experimental values found at the limits of the Ag–O and Ag–C ranges.

### 3.3. Solution Behaviour of {Ag[NHC^Mes,R^]}_n_ Compounds

The above-mentioned ESI-MS data for the **2** complexes can be rationalised by considering the break-up of the polymers into mononuclear fragments during ionisation. So, it could be assumed that, in the solution, smaller units were also probably formed from the polymers. In order to address this question, diffusion NMR measurements have been carried out. Diffusion-ordered NMR spectroscopy (DOSY) has been widely used to investigate disaggregation phenomena in solutions through the determination of diffusion coefficient (D) [58]. With this aim in mind, DOSY experiments were performed for a 1:1 mixture of imidazolium precursor **1a** and complex {Ag[NHC^Mes,H^]}_n_, **2a** (see Appendix A). From these experiments, the diffusion coefficient of **1a** was calculated as D = 7.58 × 10^−10^ m^2^/s, which was significatively higher than that observed for complex **2a** (D = 4.57 × 10^−10^ m^2^/s). This result does not support the presence of high-nuclearity species of the coordination polymers **2a** in the solution. Taking this finding, together with the ESI–MS data discussed above, it may be concluded that the polymeric structure observed in the solid state for complex **2a** is disassembled in the solution and exists, at most, as small oligomers.

From the obtained diffusion parameters, hydrodynamic radii of 5.3 and 8.8 Å for **1a** and **2a** were respectively calculated using the Stokes–Einstein equation [59]. From the optimised structures of **1a** and a dimer of **2a** (see below), it is possible to determine the maximum distance between the two farthest atoms (see Appendix A). Values of 10.8 and 18.7 Å compare well with the hydrodynamic diameters of 10.6 and 17.6 calculated for *in-solution*
**1a** and **2a**, respectively.

### 3.4. Antimicrobial Properties: Determination of Minimal Inhibitory Growth Concentrations (MIC) and Minimal Bactericidal Concentrations (MBC)

The screening of the antibacterial activity of the **2** complexes was performed using MIC and MBC measurements. Compared to Gram-negative bacteria, the complexes did not show significant biocide activity against Gram-positive strains *S. aureus* or *S. pseudintermedius* (not shown), possibly due to the difference between their cell wall structures. Therefore, the results focused on Gram-negative *E. coli* and *P. aeruginosa* and are shown in Table 1, where the MIC and MBC values for AgNO_3_, a well-known antiseptic against Gram-negative bacteria [60,61,62], are also included for an appropriate comparison under the same experimental conditions. The antimicrobial activities of ligand precursors **1** were also evaluated (not shown) and did not show antimicrobial activity towards the analysed bacterial strains. The content of silver (referring to the weight of the complex) for compounds **2a**, **2b**–**2b’**, **2c**, and **2d** are 31, 30, 28, and 27% respectively, which means they are in a comparable range. When comparing the MIC and MBC values of the **2** complexes with those obtained for AgNO_3_, only **2b’** exhibited lower MIC and MBC values than AgNO_3_ in *E. coli* assays. Concerning the *P. aeruginosa* strains, **2a** and **2b’** showed lower MIC values than AgNO_3_, although only the **2b’** complex had a lower MBC. Therefore, for both the *E. coli* and *P. aeruginosa* strains, complex **2b’** was the most effective antimicrobial agent. Importantly, these results evidence a clear difference in antimicrobial activity between the enantiomeric pair of complexes **2b** and **2b’**, which is almost certainly related to the mechanism of action of these compounds. For both bacteria, the eutomer was **2b’** complex, which was prepared with the precursor ligand **1b’** from the non-proteinaceous α-amino acid *D*-alanine. Interestingly, we have recently found similar conclusions in the antimicrobial study of α-amino-acid-based homochiral imidazolium-dicarboxylate silver(I) compounds, in which those based on *D*-amino acids showed higher antibacterial activities compared to those of the corresponding *L*-amino acids [36]. It is widely established that stereochemistry plays an important role in antimicrobial performance, and biological systems generally prefer a specific enantiomer. Thus, recent findings showed that *D*-amino acids have regulatory roles in bacteria (i.e., they can inhibit bacterial growth by affecting peptidoglycan metabolism in bacteria cells) [63]. Alterations in *D*-alanine lead to decreased cell stiffness and enhanced sensitivity to osmotic pressure [64]. Furthermore, as can be taken from the comparison of the MIC and MBC values of the **2a**–**d** complexes, which were prepared with the precursor ligands **1a**–**d** from the natural α-amino acid glycine, *L*-alanine, *L*-valine, and *L*-leucine, respectively, the nature of the R alkyl group in the NHC ligand clearly influences antimicrobial activity. Therefore, it was observed that the bulkier the R group, the less antimicrobial activity the complex has. This clear structure–antimicrobial effect relationship cannot be simply attributed to its decreasing solubility in water, because **2b’** showed more effective activities than its enantiomer, **2b**, despite these complexes having identical solubilities in water.

### 3.5. Antimicrobial Properties: Determination of Antioxidant Enzymes and Thiobarbituric Acid Reactive Substances (TBARs)

The determination of the antioxidant enzymes and reactive thiobarbituric acid substances (TBARs) of complexes **2b** and **2c** was carried out to elucidate the antimicrobial mechanism of both *E. coli* and *P. aeruginosa*. The results collected in Table 2 show different detoxification mechanisms against oxidative stress (ROS) manifested in both bacteria. Detoxification is mediated by the catalase enzyme in *E. coli*, while in *P. aeruginosa*, it is mainly mediated by superoxide dismutase, which shows a strong activity of 6 to 20 times the control. Furthermore, the quantification of malondialdehyde (MDA), which is considered as an indication of lipid peroxidation and membrane damage due to ROS [65], showed that for both complexes, MDA levels at MIC and MBC were higher than the control in both the *E. coli* and *P. aeruginosa* assays, revealing the peroxidation of membrane lipids in a dose-dependent manner (Appendix A).

### 3.6. Antimicrobial Properties’ Effects on Biofilm Formation: Evaluation by Colorimetric Technique and by Scanning Electron Microscopy (SEM)

The term biofilm has been used to denote an immobile bacterial community that adheres to a solid surface and is surrounded by an extracellular matrix typically composed of polysaccharides [40,66]. Bacterial infections associated with biofilms are extremely difficult to eliminate, as extracellular matrices protect bacteria from the antibacterial agents and the host’s immune system. Therefore, the discovery of new agents to inhibit biofilm formation and/or destroy established biofilms is attracting growing attention [66]. The effect of complexes **2b** and **2c** on the biofilm formation of *E. coli* and *P. aeruginosa* was evaluated using two different techniques: the colorimetric method [52] and SEM, and the results are presented in Table 3. Complex **2b** prevents biofilm formation at concentrations lower than MIC in both the *E. coli* and *P. aeruginosa* assays. However, the assay with **2c** showed that, in both strains, the biofilm formation was affected at MIC and not before. This behaviour again reveals the structure–antimicrobial effect relationship. The SEM technique was additionally used to evaluate the formation of biofilms on a glass surface. Appendix A for **2b** and **2c**, respectively, show the effective activity of both complexes in preventing bacterial growth and biofilm formation. As previously discussed, much attention is being directed to screening for anti-biofilm agents. The mechanisms by which these substances inhibit biofilm formation could be related to quorum sensing, signal molecules, and inhibition of DNA and nucleotide biosynthesis [67]. In our case, it could also be related to the structure or stability of the cell wall, since at MIC and MBC values, the deformation of and damage to the bacterial walls were clearly appreciated.

### 3.7. Computational Studies

The bonding capabilities of closely related chiral imidazole–carboxylate and imidazolium–carboxylate compounds were previously investigated by us from a theoretical point of view [30,31,32]. We decided to extend the theoretical study to the precursors [Im^Mes,R^], **1a**–**d**, their carbene isomers, **1′**, and the [NHC^Mes,R^]^−^ ligands of complexes **2****a**–**d**. Optimised structures of compounds **1** and **1′** are shown in Appendix A. The relative energies of carbene isomers **1′** with respect to the stable compounds [Im^Mes,R^], **1**, are collected in Table 4. In general, the carbene form is destabilised by around 9 kcal/mol, and this energy difference is substantially lower than that previously calculated for the imidazolium–dicarboxylate compound (*S*,*S*)-HL*^i^*^Pr^ (26.5 kcal·mol^−1^ [31]). This suggests that the presence of the aryl substituent in compounds [Im^Mes,R^], **1**, clearly affects such an energy difference. To evaluate the effect of the aryl group, we expanded the analysis to several *p*-substituted phenyl derivatives, [Im^Ar’,Me^] (Ar’ = *p*-XC_6_H_4_, for X = NH_2_, OH, Me, H, F, and Cl). The results obtained for this energy difference are also collected in Table 4. Electron-attractor *p*-substituents decrease the destabilisation of the carbene isomer, while an increase in the electron donation produces a major destabilisation (10.7 kcal·mol^−1^ for X = NH_2_). In fact, a clear relationship (R^2^ = 0.9795) is found between the energy difference and the Hammett parameter of the *para* substituent of the phenyl group (see Figure 2). As expected, stabilisation of the carbene form of [Im^Ar’,Me^] is accompanied by stabilisation of the carbene lone pair, which is the HOMO of the molecule (Appendix A). This fact implies a worse donation capability of the carbene ligand when it coordinates with a transition metal. The mesityl group in **1** generates the adequate electron donation to the imidazole ring to have a precursor of the carbene ligand with good donor properties for a strong interaction with the silver ion in the **2** complexes.

To simulate the polymeric nature of these complexes, head-to-tail dimeric models **2a_c_**–**2d_c_** (**c** subindex for calculated) were optimised. Figure 3 shows the calculated complex **2a_c_**, while complexes **2b_c_**–**2d_c_** are depicted in Appendix A. The description of the Ag–O and Ag–C bond distances in model **2a_c_** (2.097 and 2.104 Å, respectively) is in perfect agreement with the experimental values for **2a**, confirming the suitability of the model choice. Other structural parameters fit reasonably well with the experimental data (for instance, the C–Ag–O angle of 177.7°).

To examine the bonding capabilities of [NHC^Mes,R^]^−^ anions in **2**
*versus* the silver ion, free ligands were optimised (Appendix A) and their MOs were analysed. Figure 4 shows selected MOs for the [NHC^Mes,H^]^−^ anion, while MOs for the other anions are depicted in Appendix A. HOMO and HOMO-1 are carboxylate-centred orbitals that constitute the *in-plane* lone pairs of the oxygen atoms of this group, and are involved in the O-coordination with silver through σ-donation. HOMO-3 is a carbene-centred orbital and is the lone pair responsible for the C-coordination with silver through σ-donation.

## 4. Conclusions

Complexes {Ag[NHC^Mes,R^]}_n_, **2**, were obtained straightforwardly by reactions between Ag_2_O and imidazolium precursors [Im^Mes,R^], **1**, and they were spectroscopically and analytically characterised. The structure of **2a** is a monodimensional coordination polymer in which the silver ion is coordinated with the carbene functionality and the carboxylate group of a symmetry-related Ag[NHC^Mes,R^] unit. The silver coordination is well described by means of DFT calculations on the dimeric model {Ag[NHC^Mes,H^]}_2_. The study of the antimicrobial properties of these silver complexes was focused on Gram-negative bacteria *E. coli* and *P. aeruginosa*. From the observed MIC and MBC values, complex **2b’** showed the best antimicrobial properties, which were significantly better than those of its enantiomeric derivative **2b**. Furthermore, the comparison of the MIC and MBC values of complexes **2a**–**d** showed that the bulkier the R alkyl group, the less antimicrobial activity the complex exhibits. These facts reveal a clear structure–antimicrobial effect relationship, in which the nature of the R alkyl group in the NHC ligand clearly induces the antimicrobial activity. Further studies are in progress to analyse the action mechanism of these complexes and to evaluate the biological activities of chemically related silver complexes as therapeutic agents.

## Data Availability

Not applicable.

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
