# Peer review of "Antimicrobial Properties of Amino-Acid-Derived N-Heterocyclic Carbene Silver Complexes"

_pharmaceutics, 2022, doi:10.3390/pharmaceutics14040748_

Round 1
Reviewer 1 Report
This manuscript discusses the synthesis of amino acid-derived N-heterocyclic
carbene silver complexes and assessment of their antimicrobial activity. It was found that the R alkyl group in the N-heterocyclic carbene ligand clearly affects the antimicrobial activity.
The following questions and comments emerged when reading the paper:
- Considering that silver is one of the antimicrobial components in the synthesized N-heterocyclic carbene silver complexes. Can the authors show the content of silver in the different complexes?
- Although incorporating silver to N-heterocyclic carbene silver complexes, the antimicrobial effects of those complexes are likely dictated by released silver ions. The authors are not however measuring any release of silver ions, to correlate this with antimicrobial activity.
- For antimicrobial activity assessment, MIC and MBC were determined and in parallel, mechanisms of toxicity were assessed. However, for toxicity mechanisms only oxidative stress-related indicators are studied. Why have the authors decided to study only those indicators? Also, the oxidative stress-related enzymes and TBARS were studied only at MIC and MBC concentrations, i.e., at concentrations where most of the cells were already dead. This would probably affect the cellular responses and may by itself cause oxidative damage.
- The authors claim that silver complexes may aid in inhibition of biofilm formation. However, the results of this study are not directly supporting biofilm inhibition or destruction. The experiments of this study only show that during the constant exposure of silver complexes, E. coli and P. aeruginosa are not able to grow biofilms, thus, there is no evidence on destruction of already formed biofilms by silver complexes or inhibition of biofilm formation by bacteria after exposure to silver compounds.
Finally, please check the details: what is CMI and CMB mentioned in several places in the manuscript (e.g., on Figures S4 and S5).
Author Response
Authors would like to thank the reviewers for providing insightful and important comments on
our manuscript. As requested, we have considered all reviewers’ comments (R.C.) and
following is the list of our responses (A.R.). We have highlighted all changes in the text,
clarifying comments in blue.
Reviewer 1:
This manuscript discusses the synthesis of amino acid-derived N-heterocyclic
carbene silver complexes and assessment of their antimicrobial activity. It was found that the
R alkyl group in the N-heterocyclic carbene ligand clearly affects the antimicrobial activity.
The following questions and comments emerged when reading the paper:
1-R.C. Considering that silver is one of the antimicrobial components in the synthesized Nheterocyclic carbene silver complexes. Can the authors show the content of silver in the
different complexes?
AR. The authors would like to thank the reviewer for this interesting observation concerning
prepared complexes and have included in the text….`` The content of silver (referred to weight
of the complex) for compounds 2a, 2b´,2c´, and 2d are 31, 30, 28 and 27 % respectively,
which means they are in a comparable range´´….in 3.4 section
2-R.C. For antimicrobial activity assessment, MIC and MBC were determined and in parallel,
mechanisms of toxicity were assessed. However, for toxicity mechanisms only oxidative
stress-related indicators are studied. Why have the authors decided to study only those
indicators? Also, the oxidative stress-related enzymes and TBARS were studied only at MIC
and MBC concentrations, i.e., at concentrations where most of the cells were already dead.
This would probably affect the cellular responses and may by itself cause oxidative damage.
AR. The authors thank the reviewer and agree that other interesting mechanisms could be
studied related to toxicity. However, this is a first step multidisciplinary study in which the main
aim was to highlight the chemical preparation of new organometallic structures (synthesis,
exhaustive characterization that includes crystallographic and DFT studies among others) to
be tested as potential antimicrobial new entities, including MIC and MBC, and complemented
with oxidative stress-related indicators as a proof of one of the possible mechanisms of action
that could take part. The antioxidant enzymes have been determined since oxidative stress is
the main mechanism by which metal-derived compounds affect bacterial cells. These
enzymes (catalase, peroxidases and superoxide dismutase) are considered as markers of
oxidative stress. Regarding MDA, it is also considered as a marker of membrane lipid
peroxidation and damage to membranes, which is another mechanism by which metals
damage cells, leading to leakage. Additional mechanisms of metal toxicity include substitution
of metals in the active centers of enzymes, binging to enzymes and DNA, etc., but the later
ones have not been investigated. Concerning the concentration of compounds (MIC and MBC)
it has to be taken into account that cells were first grown in the absence of the compounds for
24 hours (enough time to get a good bacterial density) and then, they were treated with the
compounds at either MIC or MBC for additional 24h. In this sense, the bacterial growth had
already occurred before the effect of the treatment and the experiment tries to determine the
degree of cell oxidative stress and membrane damage. This is a starting step for further
applications in which other in vitro compelling studies, including cell viability, will be explored.
3-R.C. Although incorporating silver to N-heterocyclic carbene silver complexes, the
antimicrobial effects of those complexes are likely dictated by released silver ions. The authors
are not however measuring any release of silver ions, to correlate this with antimicrobial
activity.
AR. In this work our aim has been focused on demonstrating that there is an antimicrobial
behavior for different entities based on silver present and this has been proved with the
experiments included. Authors agree with the reviewer and understand these concerns in the
sense that the antimicrobial activity of silver is dependent on the silver cation Ag
+
release,
which binds strongly to biological molecules containing sulphur, oxygen, or nitrogen
heteroatoms and in some cases, it could bind to 38 different proteins. However, in this study
the Ag
+
release profile has not been included to demonstrate a sustainable release, since we
consider this data is only relevant in cases in which the silver is embedded in a polymer or
similar matrix, such as the investigations carried out in publications: a) DOI:
10.1166/jnn.2019.16663 or b) DOI:10.1002/ejic.201800640. In complexes 2, NMR and DOSY
studies do not evidence silver dissociation in solution. In this sense, the authors have also
investigated the role of Ag as antimicrobial in a recent publication to elucidate its mechanism
of action as a potential therapeutic agent (Homochiral imidazolium-based dicarboxylate
silver(I) compounds: synthesis, characterization, and antimicrobial properties. Published in
Dalton Transactions. https://doi.org/10.1039/D1DT04213K) and in these silver carboxylate
complexes there was observed silver dissociation in solution.
4-R.C. The authors claim that silver complexes may aid in inhibition of biofilm formation.
However, the results of this study are not directly supporting biofilm inhibition or destruction.
The experiments of this study only show that during the constant exposure of silver complexes,
E. coli and P. aeruginosa are not able to grow biofilms, thus, there is no evidence on
destruction of already formed biofilms by silver complexes or inhibition of biofilm formation by
bacteria after exposure to silver compounds.
AR. The authors would like to thank reviewer for this valuable comment and agree to change
this observation in the manuscript
..``Complex 2b prevent biofilm formation at concentrations lower than MIC in both E. coli and
P. aeruginosa assays´´..
..``Figs. S6 and S7 for 2b and 2c, respectively, show an effective activity of both complexes
in preventing bacterial growth and biofilm formation´´.
5-R.C. Finally, please check the details: what is CMI and CMB mentioned in several places in
the manuscript (e.g., on Figures S4 and S5).
AR. The authors acknowledge this comment and admit this a wrong terminology and have
corrected these terms in all manuscript (line 467 and figures mentioned)

Reviewer 2 Report
It was known that Silver(I)-N-Heterocyclic Carbene (Ag(Ⅰ)-NHC) complexes have shown strong antimicrobial properties against many Gram-negative and Gram-positive bacterial strains and shown very low genotoxicity. In this manuscript, Alcudia and his colleagues employed α-amino acids as ligands to generate bioinorganic metal systems with antimicrobial activities, particularly exampled by imidazole-type carboxylate compounds. As the first example of NHC-Ag-OC(O)R complexes, 2a is of interest to explain its polymeric nature.
The antimicrobial activity of several Ag(I)-NHC complexes obtained from α-amino acids-based imidazolium-monocarboxylate precursors was explored. Analysis of MIC and MBC values of these complexes revealed that the eutomer compound 2b’ showed better antimicrobial properties than those of its enanti-24 omeric derivative 2b (distomer). Additionally, the preliminary structure-antimicrobial effect relationship revealed the antimicrobial activity decreases when the steric properties of the R alkyl group in {Ag[NHCMes,R]}n increases.
The reviewer suggests the acceptance of the manuscript with minor revision. However, some questions need to be addressed as shown below:
- The manuscript needs to be polished carefully regarding the use of special characters, such as “α-amino” in line 418.
- The authors conducted DFT calculations, is there any relationship between the bonding capabilities and the antimicrobial activity? If so, please add some discussion on this point.
- S6 and S7 are probably one of the direct pieces of evidence of the compounds to irradiate the growth of bacteria. It’s better to move them into the main text, but better pictures for E. coli are highly recommended.
Author Response
Authors would like to thank the reviewers for providing insightful and important comments on our manuscript. As requested, we have considered all reviewers’ comments (R.C.) and following is the list of our responses (A.R.). We havehighlighted all changes in the text, clarifying comments in blue.
Reviewer 2:
It was known that Silver(I)-N-Heterocyclic Carbene (Ag(Ⅰ)-NHC) complexes have shown
strong antimicrobial properties against many Gram-negative and Gram-positive bacterial
strains and shown very low genotoxicity. In this manuscript, Alcudia and his colleagues
employed α-amino acids as ligands to generate bioinorganic metal systems with antimicrobial activities, particularly exampled by imidazole-type carboxylate compounds. As the first example of NHC-Ag-OC(O)R complexes, 2a is of interest to explain its polymeric nature. The antimicrobial activity of several Ag(I)-NHC complexes obtained from α-amino acids-based imidazolium-monocarboxylate precursors was explored. Analysis of MIC and MBC values of these complexes revealed that the eutomer compound 2b’ showed better antimicrobial
properties than those of its enanti-24 omeric derivative 2b (distomer). Additionally, the preliminary structure-antimicrobial effect relationship revealed the antimicrobial activity decreases when the steric properties of the R alkyl group in {Ag[NHCMes,R
]}n increases. The reviewer suggests the acceptance of the manuscript with minor revision. However, some questions need to be addressed as shown below:
1-R.C. The manuscript needs to be polished carefully regarding the use of special characters, such as “α-amino” in line 418.
AR. The authors thank reviewer for this observation and have changed it.
2-R.C. The authors conducted DFT calculations, is there any relationship between the bonding capabilities and the antimicrobial activity? If so, please add some discussion on this point.
AR. The authors acknowledge this comment but, in this work, the DFT tool was employed to analyze the bonding capabilities of NHC ligands, to prove the suitability of the [NHCMes,R] ligands for strongly coordinating silver cation and to simulate the solution behavior of complexes 2. For these reasons, there is not a direct relationship between the DFT analysis and antimicrobial activity.
3-R.C. S6 and S7 are probably one of the direct pieces of evidence of the compounds to
irradiate the growth of bacteria. It’s better to move them into the main text, but better pictures for E. coli are highly recommended.
AR. The authors agree with the reviewer that these data could eventually be added to the mail text, but unfortunately are very sorry since we do not have a better-quality one, and for this reason think is better to locate them where they originally were. However, despite the quality, the experiment showed that these compounds inhibit growth of bacteria
